# Vimentin Cytoskeleton Architecture Analysis on Polylactide and Polyhydroxyoctanoate Substrates for Cell Culturing

**DOI:** 10.3390/ijms22136821

**Published:** 2021-06-25

**Authors:** Karolina Feliksiak, Daria Solarz, Maciej Guzik, Aneta Zima, Zenon Rajfur, Tomasz Witko

**Affiliations:** 1Faculty of Physics, Astronomy and Applied Computer Science, Jagiellonian University, 30-348 Cracow, Poland; karolina.feliksiak89@gmail.com (K.F.); daria.solarz@doctoral.uj.edu.pl (D.S.); 2Jerzy Haber Institute of Catalysis and Surface Chemistry, Polish Academy of Sciences, 30-239 Cracow, Poland; ncguzik@cyfronet.pl; 3Faculty of Materials Science and Ceramics, AGH University of Science and Technology, 30-059 Cracow, Poland; azima@agh.edu.pl; 4Jagiellonian Center of Biomedical Imaging, Jagiellonian University, 30-348 Cracow, Poland

**Keywords:** polyhydroxyalkanoates (PHO), polylactide (PLA), biomaterials, MEF cells, cytoskeleton, vimentin, confocal microscopy

## Abstract

Polylactide (PLA), widely used in bioengineering and medicine, gained popularity due to its biocompatibility and biodegradability. Natural origin and eco-friendly background encourage the search of novel materials with such features, such as polyhydroxyoctanoate (P(3HO)), a polyester of bacterial origin. Physicochemical features of both P(3HO) and PLA have an impact on cellular response 32, i.e., adhesion, migration, and cell morphology, based on the signaling and changes in the architecture of the three cytoskeletal networks: microfilaments (F-actin), microtubules, and intermediate filaments (IF). To investigate the role of IF in the cellular response to the substrate, we focused on vimentin intermediate filaments (VIFs), present in mouse embryonic fibroblast cells (MEF). VIFs maintain cell integrity and protect it from external mechanical stress, and also take part in the transmission of signals from the exterior of the cell to its inner organelles, which is under constant investigation. Physiochemical properties of a substrate have an impact on cells’ morphology, and thus on cytoskeleton network signaling and assembly. In this work, we show how PLA and P(3HO) crystallinity and hydrophilicity influence VIFs, and we identify that two different types of vimentin cytoskeleton architecture: network “classic” and “nutshell-like” are expressed by MEFs in different numbers of cells depending on substrate features.

## 1. Introduction

Polyhydroxyoctanoate (P(3HO)) and polylactic acid (PLA) are biocompatible and biodegradable polymers with a high application potential in tissue regeneration, medicine, and cellular studies [1,2]. While PLA is very deeply researched and broadly used in modern medical solutions (i.e., as implants, drug carriers, or scaffolds for tissue engineering) [3], P(3HO) is a quite novel material, the applications of which are currently being developed. It is worth noting that both PLA and P(3HO) are thermoplastic, fully biodegradable, and can be obtained from monomers of natural origin. However, despite the wide use of PLA in modern medical applications, it also has some limitations. Therefore, we consider P(3HO) as a potential replacement of PLA, especially where the values of Young modulus or durability exceed PLA’s adaptability. The need for use of hydroxy acid-based polyesters in biomedicine has grown, since from an environmental point of view, these polymers are a reliable alternative to the use of traditional, petroleum-based plastics that increase pollution and global warming [2,4,5]. Moreover, PLA monomer (lactic acid) can be obtained from the bacterial fermentation of renewable sources, such as sugar or corn starch, and P(3HO) is synthesized by bacteria directly from bioresources.

### 1.1. PLA

PLA, a linear, aliphatic, thermoplastic polyester, is recognized as one of the most innovative materials and is being employed in many industrial applications [6]. Especially, the eco-friendly advantages of PLA make it an attractive choice in current applications—it is recyclable and compostable, its production consumes carbon dioxide, and it is also biocompatible and non-toxic [2]. The key determinants of PLA’s success in bioengineering are its biodegradability and the direct effects of PLA on cell behavior, morphology, and differentiation [7,8]. The topography of the PLA can affect all levels of cell–substrate, cell–cell, and also cell–ECM (extracellular matrix) interactions [9]. Important parameters of PLA which can exert influence on cellular behavior are its elasticity (Young’s modulus > 3.5 (×10^3^) MPa, friction coefficient 0.1–0.16) and hardness, 7–11 MPa [1,10,11].

### 1.2. P(3HO)

Polyhydroxyalkanoates (PHAs) are a group of the novel, optically active, biodegradable polyesters, sourced from bacterial synthesis, that have become a popular and interesting alternative for currently used biopolymers, in terms of applications in the medical field. PHAs are considered attractive due to their excellent biological features—full biocompatibility and lack of toxicity to mammalian cells and tissues—all due to the fact that the monomers of PHAs ((*R*)-3-hydroxylated fatty acids) are chemically identical to the products of β-oxidation—the standard cell metabolic process [12]. Their other advantage is an eco-friendly background, as they are collected in the form of granules in the microbes’ cells, serving as internal carbon and energy storage components [13]. One of the materials recently gaining interest among the PHA group is polyhydroxyoctanoate—P(3HO). P(3HO) is already known for its numerous advantages in terms of biocompatibility and biodegradability. Due to its beneficial elasticity (Young’s modulus 11.4–34 MPa) [4,14], endurance (friction coefficient 0.198–0.202), and low fragility (hardness 3.2–4.5 MPa) [1], it is being reviewed in terms of potential heart valve replacement, in wound-healing studies, and for joint endoprostheses [15,16,17]. Although in our previous works we ran a series of experiments describing the influence of P(3HO) on cell behavior, migration, morphology, and cytoskeleton, still there are some uncovered areas of interest on the specific substrate–cell interaction [1,4,18].

### 1.3. Substrate Features Influencing Cell Behavior

Mechanical features of the substrate are key determinants of cells’ behavior, which can be modified in terms of motility [19], proliferation [20], or differentiation [21]. Topography of the substrate has a direct influence on the morphology of the cells, which adhere to the surface of a substrate in a naturally occurring process—contact guidance—a response of the cells to structures on the micro- and nanoscale. The capability of sensing the external environment is provided by the ECM network interactions with transmembrane proteins that can transmit and transform mechanical signals, resulting in changes in the cytoskeleton. Moreover, cells can react and adjust to chemical stimuli, such as the composition of the substrate, coverage of the substrate surface with proteins, or its roughness [22]. Additionally, the elasticity of the substrate matters in terms of cellular response. Fibroblasts, for instance, tend to direct their movement towards the substrates of higher values of Young’s modulus [23]. Controlling the stiffness of the substrate, as a single characteristic, has proven to be sufficient for the differentiation of human mesenchymal stem cells into mature adipocytes or osteocytes [21]. The influence of substrates’ stiffness is most widely researched on actin cytoskeleton [24], and it is known that a higher amount of actin stress fibers is formed on substrates of higher stiffness [1,25]. Vimentin intermediate filaments also react to mechanical features, i.e., stiffness of the substrate. VIFs detergent-solubility is dependent on the substrate stiffness, and it directly affects the cell spreading and membrane ruffling [26].

### 1.4. Vimentin

Vimentin is a type III intermediate filament protein of 57 kDa molecular weight, consisting of 466 amino acids. Its rod domain, containing 310 amino acids, is α-helical, with the N-terminal head domain of a 102 amino acid-long sequence, and a C-terminal hydrophobic tail on the other end.

Vimentin architecture in cells is dynamic and can be identified at the same time as non-filamentous particles of ULF, short filaments of ULFs (or squiggles) [27], and mature, long fibers. Filaments reorganize constantly, in response to the changes of the shape and location of the cells, however rearrangement of vimentin filaments is not reliant on complete disassembly, such as in the polymerization of actin or microtubules.

The structure of vimentin IFs changes during the cell’s crucial processes, such as migration, cell cycle, or cell proliferation, where it can assemble in all three forms of the filaments—either particles of ULFs, short ULF filaments (squiggles), or long fibers [28,29]. Long, mature vimentin fibers create a network that spans the cell, linking the cell edges with the nucleus. The network surrounds the nucleus and spreads to the periphery of the cell, where many more soluble, short fibers can be identified [30]. In MEF cells, complex networks of vimentin fibers can be identified mostly in the perinuclear area and tail. Long VIF can also be located in the regions between the nucleus and the lamella, but the number of these drops, and they terminate at the proximal region of the lamella. Here, the increased number of short filaments (i.e., squiggles) appears, together with non-filamentous vimentin precursor particles. Squiggles’ number drops in the distal lamella regions, where the number of particles increases. Finally, the particles are most abundant in the leading edge of lamellipodia. In conclusion, the forms of vimentin within one cell present a gradient of assembly, with the highest concentration of particles in lamellipodia, and the highest number of vimentin long fibers located in the perinuclear region and tail [31].

Figure 1 presents the three basic substructures of vimentin within a single cell. Fibers—present in the perinuclear area and tail—gradually convert into shorter filaments (i.e., squiggles), eventually becoming particles of vimentin at a certain distance from the nucleus. These substructures within the area of a single cell are just one way of distinguishing the variety of vimentin. We found, however, that the vimentin network can be assembled differently at the same time for cells cultured on different substrates. The architecture of the whole network can be classified into a few clearly distinctive types, that differ in terms of distribution, compaction, and coexistence of vimentin fibers. Since vimentin has such differences in terms of assembly for single-migrating cells, the next question is “what is the vimentin architecture in a monolayer of a fibroblast?”, which is in turn connected with the wound-healing process.

Since P(3HO) and PLA are proven to have the potential to be applied in wound management [10,32], and we have already determined the mechanical features of these materials [1,4,18], we decided to investigate the influence of these substrates on vimentin intermediate filaments, which are also known to be a key factor in the wound-healing process [33]. This study aims to evaluate the impact of biodegradable polymeric materials, i.e., PLA and P(3HO), on the conformations of vimentin network after culturing mouse embryonic fibroblast cells (MEF). The studies we performed were focused on providing an insight into the detailed morphology of the vimentin cytoskeleton, and confronting that with the predictions of VIFs behavior and assembly, based on the current state of knowledge of the cytoskeletal architecture on different substrates used for cell culture. Here, the diverse vimentin cytoskeleton architecture is thoroughly examined.

## 2. Results

### 2.1. Nature of PLA and PHO Growth Substrates

The physicochemical properties of PLA and PHO films were thoroughly studied and characterized in our previous works [1,4,18], however for the purposes of this work, we performed polarized-light observations of the substrates, to present the irregular structures on the surface of PLA and PHO films. The observations were conducted with different magnifications.

For the PLA film in the 5× and 10× magnification images (Figure 2A,B), some irregularly distributed thickenings or rough bulges can be noticed. In the 40× magnification image (Figure 2C), these thickenings can be observed as fairly rough protuberances of different sizes (i.e., 10–400 μm width) distributed throughout the remaining film area, which is also rough in the nanoscale. Topography images of PHO films (Figure 2D–F) showed a high number of stable nucleation seeds, quite a homogenous structure with a low number of irregularities in the z-axis, and several surface defects, i.e., microcracks. The images reveal the micro- and nanoscale roughness features, with no privileged orientation of the irregularities, which suggests that the crystallization direction was random as well as the distribution of polymeric chains. From the comparison of polarized light images of PHO and PLA (Figure 2), it can be seen that both surfaces have irregular roughness features, with the PLA roughness increased in the z-scale, unlike the irregularities on PHO films, which are flatter in the z-dimension.

### 2.2. Vimentin Network Architecture in Single Cells vs. the Features of the PLA and P(3HO) Films

The basic vimentin structure is described in the literature as an irregular network localized around the nucleus, spreading from the edges into the interior of the cell [30]. This structure type was mostly found in cells grown on glass, and on P(3HO), whereas the cells cultured on PLA were noticed to prefer the expression of a “nutshell” variant (Figure 3).

In the present work, we analyzed 288 MEF cells altogether, from which 95 on P(3HO), 80 on PLA, and 113 on glass. For 4% of the 288 cells, we were unable to identify the exact structure of the vimentin cytoskeleton.

The upper panel in Figure 3 presents the percent of the cells showing each morphology of vimentin on different substrates.

The classic network was found mostly on glass, where 83% of the cells were expressing that phenotype. For P(3HO), the number of cells with the classic network was also high—74%, whereas for PLA, that number was only 44%. Moreover, in the PLA case, we identified a higher number of cells expressing a nutshell structure—49%, which suggests the strong influence of the substrate on the configuration of the vimentin cytoskeleton. The results were tested with the *χ*^2^ test, which confirmed that it is statistically significant, meaning that the type of the vimentin cytoskeleton is dependent on the substrate used (*χ*^2^_0.01_ = 35,600, *p* < 0.01). The dotted lines in Figure 3′s upper panel present the trend of the occurrence of the vimentin subtype on each substrate. The classic network tendency, shown on the graph, is growing in terms of a number of cells with that type of vimentin skeleton, whereas the nutshell structure is trending in the opposite direction. Since in our previous works we identified changes of behavior of actin cytoskeleton depending on Young’s moduli of the substrate, we also considered that such influence is maintained for vimentin. The results suggest, however, that for the vimentin cytoskeleton subtype, factors other than the substrate’s Young’s modulus are determining the formation of VIFs.

Figure 4 presents the examples of two types of vimentin cytoskeleton for two different MEF cells grown on P(3HO) (A–D) and PLA (E–H). First of all, these two structures are different in terms of network appearance. The “classic” network (Figure 4A–D) has distinguishable VIFs, that are arranged throughout the cell body, clearly surrounding the nucleus, and closely attached to its immediate area. Our research suggests an association of that type of vimentin cytoskeleton with the higher hydrophobicity of the P(3HO) films, as well as higher crystallinity, in comparison to PLA. The other example (Figure 4E–H) presents the “nutshell” structure variant of the vimentin network, which differs from the classic version in terms of deployment throughout the cell, and with respect to the nucleus. The “nutshell” network remains distant from the nucleus on the bottom of the cell, and only attaches to the nuclear area at the top of the nucleus, covering it, and resembling a shield, or a shell of a nut. The “nutshell” structure forms a single layer and is not identifiable under the nucleus, contrary to the classic version, in which the fibers are spreading both under and above the nucleus. This “nutshell” subtype of vimentin could be associated with the higher hydrophilicity of the substrate, as well as with the lower crystallinity. That would support the proposition that the more crystalline P(3HO) films induce more points of focal adhesions for the cell, which translates into the presence of the vimentin under the nuclear area, at the bottom surface of the cell.

### 2.3. Vimentin Cytoskeleton Analysis

The vimentin cytoskeleton was analyzed with ImageJ (Fiji) ver. 1.51h software in the whole area taken-up by the vimentin within the cell, and in three separate regions of the cell—under the nucleus, in the middle of the cell height, and above the nucleus. We performed the ”whole area occupied by vimentin” approach to check if the amount of vimentin in classic vs. nutshell structure differs concerning the area occupied by the vimentin. Below, in Figure 5, we show an example of how the calculations were performed.

The cell’s image parameters were initially adjusted, and region of interest was selected on the basis of the thresholding. Finally, the maximum intensity picture in 16-bit format was measured in terms of the amount of vimentin contained within the area taken by the vimentin (percentage value).

A similar procedure was performed for the chosen cell regions: a 10 × 10 μm square was selected in the area nearest to the center of the nucleus. Three regions were measured—under the nucleus, in the middle of the height of the cell, and above the nucleus. From our observations, it turns out that for the “nutshell” structure of vimentin, there is almost no vimentin present. The same situation relates to the middle part of the cell, whereas for the classic structure, vimentin fibers are in most cases present, and both under the nucleus and in the middle of the cell height, the nutshell structure differs distinctly.

The process of selecting the regions under the nucleus, in the middle of the cell height, and above the nucleus is presented in Figure 6.

Regardless of the substrate used, the classic vimentin network occupies 91 ± 5% of the area, vs. 84 ± 10% per nutshell structure. From a statistical point of view, these values are not significantly different. As per the differentiation of the regions and checking the amount of vimentin within the specific area—checked in the 10 × 10 μm squares placed under, in the middle of the cell height, and above the nucleus—the data is as follows (Figure 7).

At a confidence level of 95%, significant differences can be noticed between the selected regions, especially under and above the nucleus in classic and nutshell structures for all substrates. Additionally, for the area above the nucleus in the classic network for P(3HO) cultured cells, the amount of vimentin is significantly lower than for PLA and glass. Nutshell structure values for all three groups of cells—cultured on PLA, P(3HO), and glass—are near 0 in the region under the nucleus and in the middle of the cell height. For the area above the nucleus, the differences are not statistically significant. Since the calculations were not performed on all 288 cells that were captured, due to the limited technical capacity and insufficient magnification of some images that disabled the thresholding of the 16-bit pictures, the results do not cover the full sample. The results, however, based on a confidence level of α = 95%, support the assumption of the minimal presence of vimentin both under the nucleus and in the middle of the cell height for the nutshell structure, as well as much lower amounts of vimentin in the middle area of the cell height for the classic structure. In that case (classic version), the presence of vimentin in the middle of the cell height, i.e., often in the center of the nucleus, is correlated with the fact that vimentin fibers create bundles, that groove the nucleus, which was broadly researched in our previous work [34].

## 3. Discussion

The spatial architecture of vimentin, as shown in the present work, is diverse in MEF cells: it can be expressed as the classic vimentin network, closely attached to the nucleus, and present throughout the cell volume, or as a “nutshell” structure, that resembles a shell or a coating, covering the cell from above, only attached to the top of the nucleus (Figure 3).

We have shown that the number of cells with vimentin in classic vs. “nutshell” structure was different for different substrates, particularly for PLA, and thus we concluded that this reflects the influence of the substrate on the morphology of the vimentin cytoskeleton.

We also identified that higher hydrophilicity of the substrate and lower crystallinity might induce the increased presence of the nutshell structure of vimentin (Figure 4). Since the crystallinity also enhances roughness, that could support the idea that the more crystalline P(3HO) films induce better fibroblast adhesion, which translates into more focal contacts of the cell with the substrate, which in turn enhances the presence of the vimentin at the bottom of the cell and under the nuclear area. Since generally better adhesion is achieved on the substrates of higher crystallinity [35], this may suggest that the classic network form of vimentin cytoskeleton is the preferred one.

To further evaluate the possible influence of the PLA and PHO films’ features on cellular behavior, we also considered the wettability and crystallinity of these two polymers, as these features are proven to have an impact on the biological performance of the materials, including cellular behavior, attachment, migration, and proliferation [36]. From the hydrophilicity assessment, performed in the previous study [18], we identified that P(3HO) was more hydrophobic than PLA. The contact angle measured for PLA was 68° ± 2° and for P(3HO)—100° ± 6°. For phase composition, that was also performed in our previous research [18], PLA films were less crystalline (XC,DSC = 4%) than semi-crystalline P(3HO) (XC,DSC = 37%). Since these two features are confirmed to influence the migration of the cells, as less hydrophobic and less crystalline PLA was connected to the lower migration speed of the single cells than more hydrophobic and semi-crystalline P(3HO), we also examined the migration process in a wound-healing model. Results showed that wound-healing processes on P(3HO) and glass were occurring with similar speed and dynamics, and for PLA, the overgrowing of the wound took much more time [18].

Since the materials of higher hydrophilicity are documented to have a beneficial influence on cellular adhesion and proliferation, and also have a better capacity for adsorption of proteins, that might also connect to the single-cell behavior in terms of vimentin structure [37]. In our previous works, we indicated that both wettability and crystalline structure of the substrate influence the cell migration speed. More hydrophilic and less crystalline PLA was connected to the lower migration speed of the single cells than more hydrophobic and semi-crystalline P(3HO) [18]. Considering these features, we postulate that the wettability alongside crystalline structure significantly influences the occurrence of the nutshell vimentin subtype in MEF cells. The nutshell and classic subtypes of vimentin cytoskeleton assemblies differ distinctly.

Due to the fact that proliferation, adhesion, and migration of the cells can be affected by the external factors (substrate characteristics), and also by the performance of actin cytoskeleton and nucleus behavior, it seems necessary to explore the interior of the cell more deeply. The questions are, how the assembly of the cytoskeleton and its separate elements correlates with the signals received from the external environment, but also, how the inner cytoskeletal filaments’ assembly influences cellular behavior on a specific substrate. Since intermediate filaments form fibers connecting the perimeter of the cell (actin cytoskeleton) and the nuclear area, they seem to be a crucial element in transferring mechanical signals from the external environment to the cell core. That is why we decided to consider vimentin’s behavior in MEF cells cultured on PLA, P(3HO), and glass.

## 4. Conclusions

P(3HO) and PLA are materials with a high potential of use in medicine, tissue engineering, and biotechnological applications. Here, we obtained thin polymer films from casting P(3HO) and PLA dissolved in ethyl acetate, that served as substrates for cell culture. The films used were proven to exert their influence on cell cytoskeleton architecture, which has been checked in MEF intermediate filament configuration changes, i.e., vimentin.

In our previous research, we focused on Young’s modulus, which correlated with the morphology of MEF cells. The elasticity module value is similar on PLA (~74 GPa) and glass (~73 GPa), and for these two substrates, we observed that MEF cells tend to behave alike. Cells cultured on PLA and glass expressed flattening in the lower area and the nucleus, and also presented visible actin stress fibers and focal adhesions. For P(3HO), for which the Young’s modulus is 3 orders of magnitude smaller (~33 MPa) than for PLA and glass, there was also a noticeable change in the cellular behavior—despite the flattening of the body of the cell, the nucleus remained round, however the amount of actin stress fibers dropped [1,18]. Scratch tests, performed for the P(3HO) and PLA films, showed an increase and decrease of friction coefficient in the local areas observed for both materials, which confirmed the differentiation of topography within the sample for both of the polymeric films [4,18]. Since the classic network is preferably expressed on P(3HO) and glass substrates, which have very different Young’s moduli values (~33 MPa—P(3HO), ~72 GPa—glass), and the nutshell-like vimentin structure is preferably expressed in cells cultured on PLA (with Young’s moduli similar to the values of glass), it can suggest that other features of the substrate have a greater impact on vimentin’s behavior than the elasticity module alone. Considering the above, we could think of a nutshell structure of vimentin as a defensive cellular response to unfavorable or undesirable environmental conditions. Although the exact reasons for having two different subtypes of vimentin structure are not fully understood or identified, there is strong proof that the changes of physiochemical, topographical, and chemical features of the substrate used for cell culturing have an impact on cellular behavior, including intermediate filaments of vimentin in MEF cells. Considering the latest discoveries on identifying vimentin as an attachment site for the SARS-CoV-2 virus [38], it strengthens the importance of further research on the cytoskeleton elements, and the analysis of the individual local behavior of intermediate filaments and further determination of their far-reaching effects. Our research, showing the influence of physicochemical features of the substrate on obtaining two distinctly different vimentin cytoskeleton structures, opens up the field for more insightful research on the issue of the external environment conditions, inducing changes not only in the architecture of intermediate filaments but also on its impact on tissue behavior and holistic effects in a whole organism.

## 5. Materials and Methods

### 5.1. Preparation of P(3HO) and PLA Substrates

P(3HO) was obtained from octanoic acid as described by Sofinska et al. [4]. *Pseudomonas putida* KT2440 fermented octanoic acid in a 5 L bioreactor in minimal slat medium to yield polyhydroxyoctanoate—P(3HO) polymer. The produced biomass, containing P(3HO) polymer, was then dried out and extracted with the use of ethyl acetate after the fermentation process ended. The purification of the solution of polymer in ethyl acetate was performed by the use of activated charcoal and a 0.2 µm filter. The precipitated P(3HO) in cold methanol was then dried. PLA used for the substrates was purchased from MG Chemicals (Burlington, ON, Canada) in the form of a 1.75 mm diameter filament for 3D printing (PLA17TL1—density of 1.24 g/cm^3^, print temperature of 180–230 °C, tensile strength of 1 MPa, and Young’s modulus of 3.5 GPa) [38].

Both P(3HO) and PLA polymers were dissolved in ethyl acetate in a proportion of 0.5 g of polymer per 10 mL of solvent, then films were cast on a glass bottom dish: 70 µL of the solution was used for casting a layer of a film per one glass bottom dish to obtain a flat and even surface, for which the thickness did not exceeded 100 µm. After that, films were left to dry for 48 h at 50 °C. Substrates were rinsed with sterile PBS twice before cells were seeded.

### 5.2. Microscopic Observations of the Surface of Polymeric P(3HO) and PLA Films

The images of the surface of PLA and P(3HO) films in polarized light microscopy were obtained with a Zeiss Axio Observer Z.1 microscope at 5-, 10-, and 40-times magnifications.

### 5.3. Cell Culturing and Staining on P(3HO), PLA, and Glass

Mouse embryonic fibroblasts (MEF) were cultured on three different substrates, under sterile conditions in an incubator that provided stable temperature of 37 °C and atmosphere with 5% of CO_2_. DMEM Low Glucose (Dulbecco’s Modified Eagle Medium, Biowest), supplemented with 10% of FBS (Fetal Bovine Serum) and 1% of antibiotics (penicillin and streptomycin, Sigma-Aldrich, St. Louis, MO, USA), was used for culturing. The cells that were used in the experiments were between the third and ninth passages, grown each time to approximately 80% of confluence. For final measurements, cells were grown for 12 h on each substrate: P(3HO), PLA, and glass for reference. Then, cells were fixed in 4% formaldehyde solution in PBS, permeabilized for 5 min with 0.1% Triton-X solution in PBS, and treated with 30% BSA (Bovine Serum Albumin) solution to block unspecific binding. After that, the overnight staining at 5 °C with Abcam Recombinant Anti-Vimentin antibody—cytoskeleton marker ab92547—in BSA solution was performed, followed by 3 h-long staining of the nucleus with DAPI dye (Thermo Fisher Scientific, Waltham, MA, USA), and secondary staining of vimentin with Alexa Fluor 633 (Thermo Fisher Scientific) at room temperature (RT).

### 5.4. Microscopic Observations of Vimentin Cytoskeleton of MEF Cells Grown on Different Substrates

The imaging of the stained cells was conducted on a Zeiss Axio Observer Z.1 microscope with a LSM 710 confocal module. The analysis of the captured images was performed with Zeiss ZEN Black version 8.1.0.484, PALMRobo version V 4.6.0.4, and Fiji app (ImageJ 1.52p) software. The imaging was performed on single cells with an oil immersion objective of 40× magnification, and a numerical aperture (NA) of 1.4. To image a single cell, the field of view was adjusted to cell density in each case.

### 5.5. Vimentin Cytoskeleton Analysis

The vimentin cytoskeleton analysis was conducted using ImageJ (Fiji) ver. 1.51h software for both entire vimentin density throughout the whole cell and distinct individual regions—under the nucleus, in the middle of the cell height, and above the nucleus.

### 5.6. Statistical Analysis

A statistical analysis employing a χ^2^ test was performed in order to verify dependencies between the classic/nutshell vimentin networks and the substrate used. The differences were deemed statistically significant at probabilities of *χ*^2^_0.01_ (Origin Pro 2019 Software, OriginLab Corporation, Northampton, MA, USA). The experiment was performed 5 times for each substrate, and 15–25 cells were scanned per sample.

## Figures and Tables

**Figure 1 ijms-22-06821-f001:**
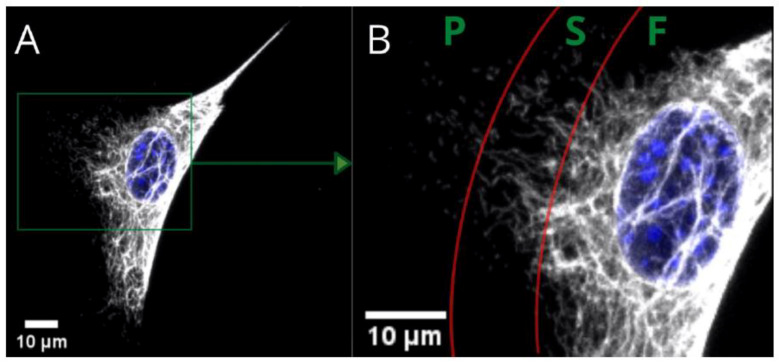
Confocal microscopy image of fibers (F), squiggles (S), and particles (P) of vimentin in a single MEF 3T3 cell. Long vimentin fibers are clearly visible above the nucleus in picture (**A**). Picture (**B**) is a magnified area that shows a gradual transition from fibers to short filaments (squiggles), and from squiggles to particles.

**Figure 2 ijms-22-06821-f002:**
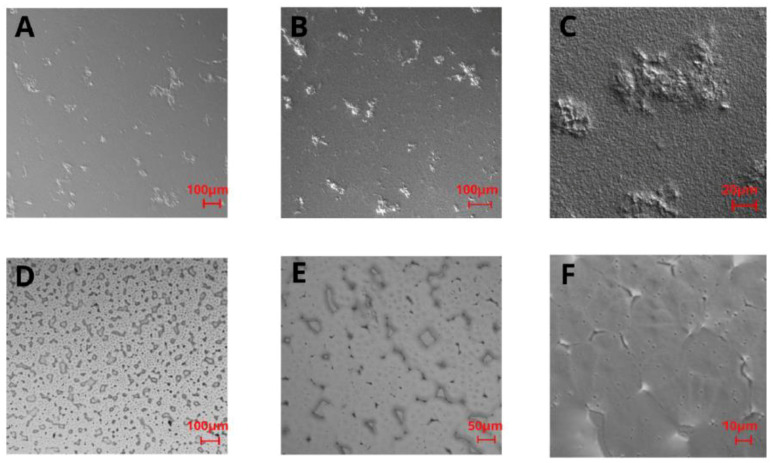
Polarized light images of PLA (**A**–**C**) and P(3HO) (**D**–**F**) films in 5× magnification (**A**,**D**), 10× magnification (**B**,**E**), and 40× magnification (**C**,**F**).

**Figure 3 ijms-22-06821-f003:**
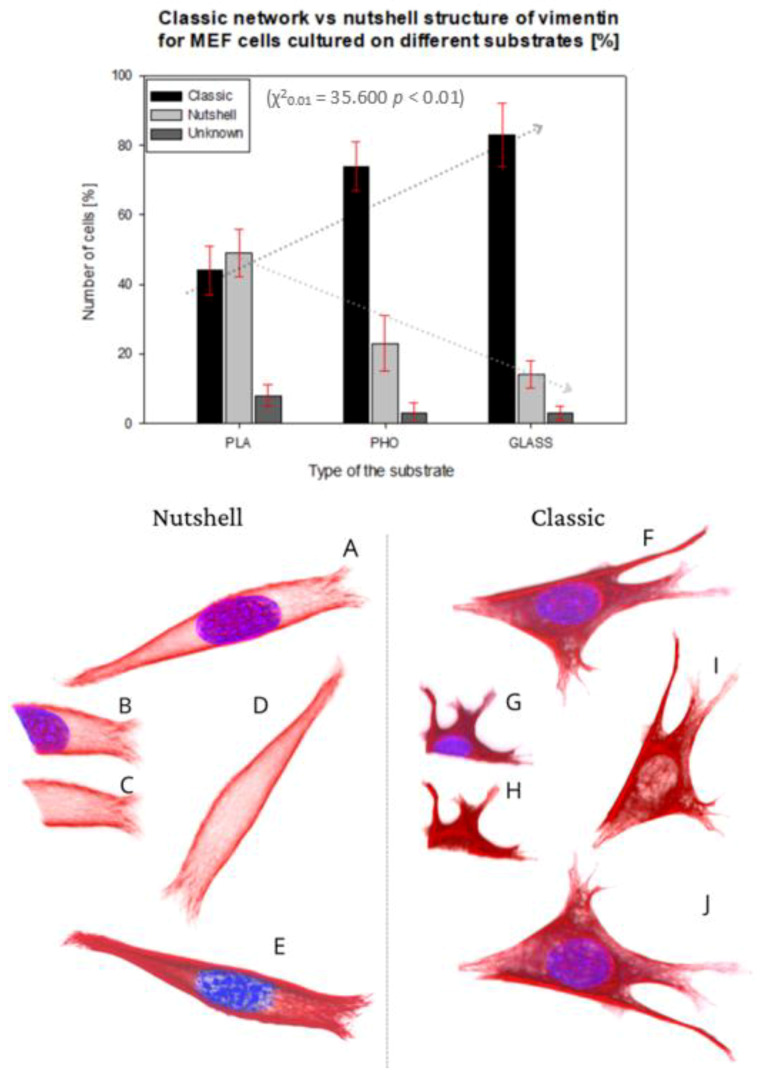
Percentage number of cells presenting classic network or nutshell structure of vimentin in cultures on PLA, P(3HO), and glass. 3D projection of examples of nutshell structure cell (**A**–**E**), and classic structure cell (**F**–**J**). Vimentin—Red, Nucleus—Blue. (**B**,**G**) Fragment of confocal image slice showing the structure of vimentin close to or in the area of the nucleus ((**C**,**H**) DAPI channel not present).

**Figure 4 ijms-22-06821-f004:**
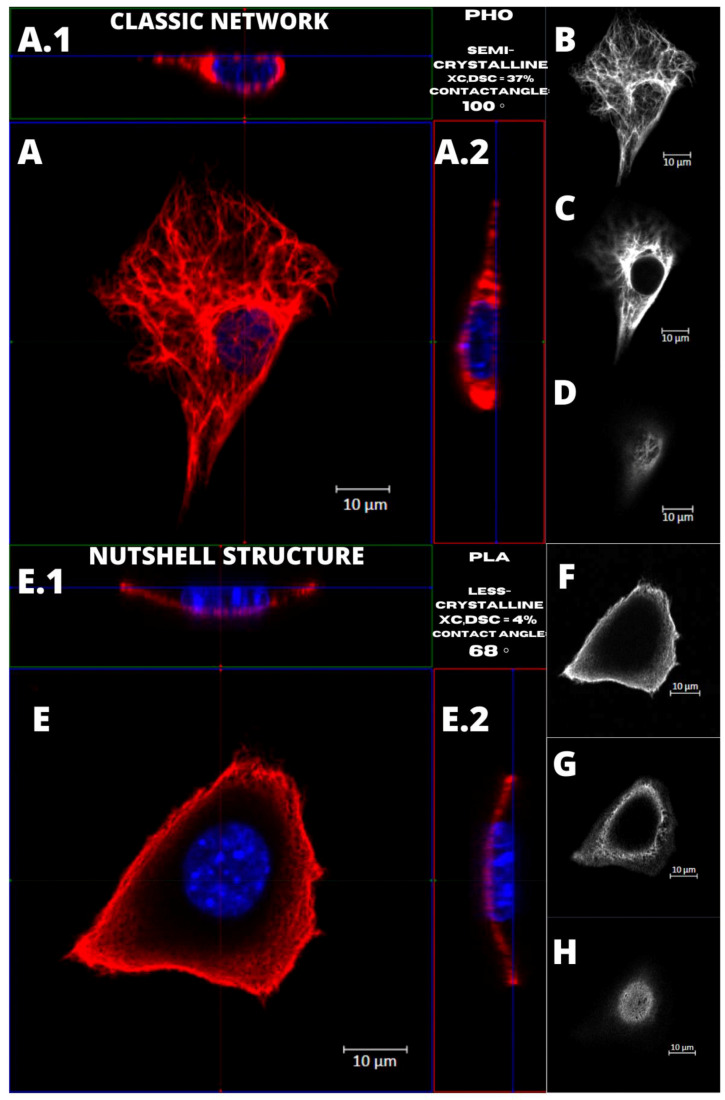
Comparison of the vimentin classic network and nutshell structure appearance. Cell with classic network cultured on P(3HO), cell with nutshell structure cultured on PLA. (**A**,**A.1**,**A.2**,**E**,**E.1**,**E.2**) Orthogonal view. (**B**,**F**) Single z-stack images of the bottom of the cell. (**C**,**G**) Single z-stack images of the middle of the cell. (**D**,**H**) Single z-stack images of the top of the cell.

**Figure 5 ijms-22-06821-f005:**
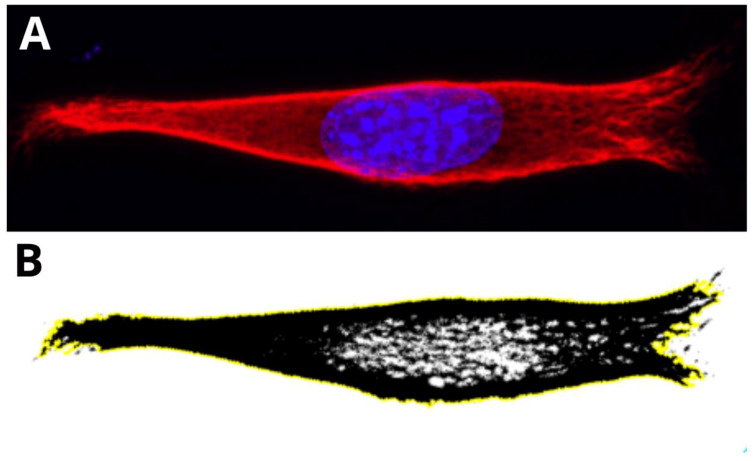
Presentation of a step performed for analysis of the vimentin cytoskeleton. Example of a nutshell structure of MEF cell cultured on PLA. (**A**) The cell is cropped, and image parameters are adjusted initially. (**B**) 16-bit (B&W) image of the vimentin cytoskeleton, after selection of ROI (region of interest—yellow line) based on color thresholding.

**Figure 6 ijms-22-06821-f006:**
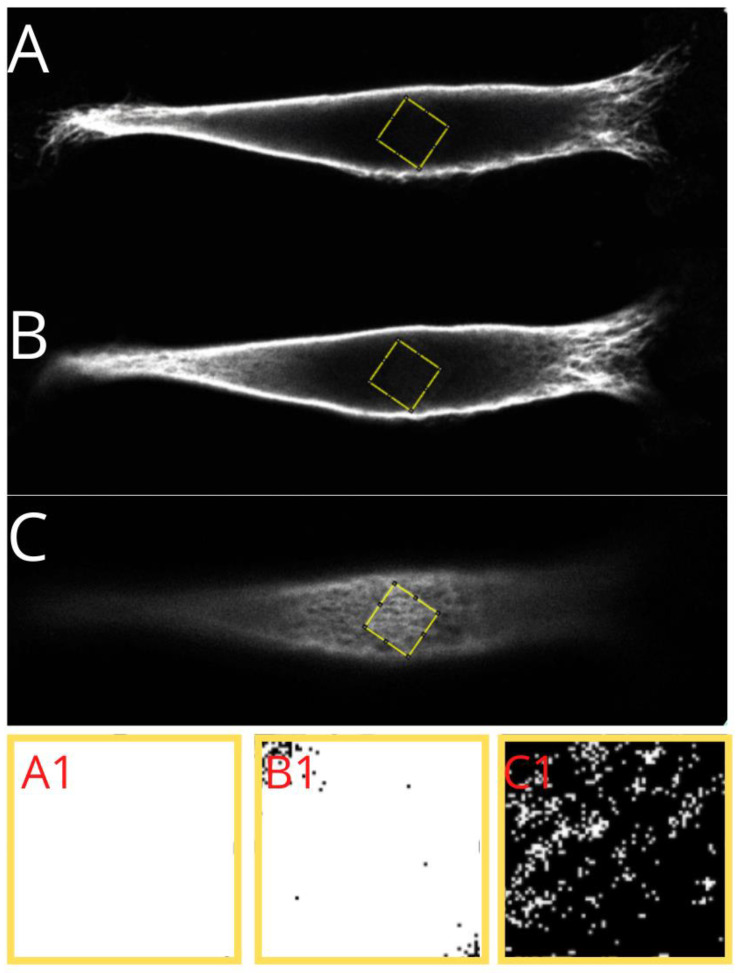
Presentation of a step performed in analysis of vimentin cytoskeleton. Identification of the regions under the nucleus (**A**), in the middle of the cell height (**B**), and above the nucleus (**C**), and the threshold areas in magnification, respectively (**A1**,**B1**,**C1**)—presenting the identified actual amount of vimentin in selected regions.

**Figure 7 ijms-22-06821-f007:**
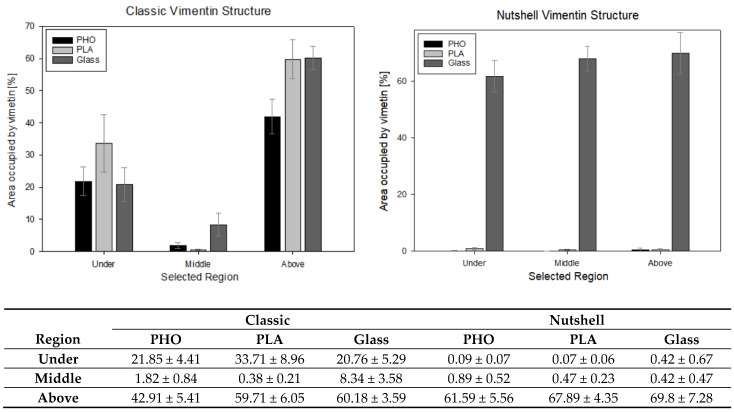
Presentation of the percentage amount of vimentin measured in 10 × 10 μm squared region of interest (ROI) in the areas under the nucleus, in the middle of the cell’s height, and above the nucleus. The table presents the calculated average values % ± standard error value.

## Data Availability

Not applicable.

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
