# Peer review of "Vimentin Cytoskeleton Architecture Analysis on Polylactide and Polyhydroxyoctanoate Substrates for Cell Culturing"

_ijms, 2021, doi:10.3390/ijms22136821_

Round 1
Reviewer 1 Report
Review to authors
This manuscript showed how certain PLA and P(3HO) features affect the Vimentin Intermediate Filaments (VIFs) and two different types of vimentin in mouse embryonic fibroblast cells were identified. I think this work is interesting and provides insight information about how biomaterials affect the VIFs. Generally, this manuscript is well-planned, which I think major revision is needed. And the detailed comments are as below:
- The language still needs to be further polished, some words mat be incorrect, like page 3 line 143, the authors double spelled the word “in”.
- More information about the correlation between the PLA and P(3HO) should be provided in the introduction part. From this version of manuscript, it’s hard to find the correlation between the PLA and P(3HO). Why the PLA was chosen to be compared with the P(3HO) ?
- More information about the mechanical features of both PLA and P(3HO) which were used in this experiment should be provided. Many different features like elasticity or hardness can affect cells behaviors. However, the mechanical features of PLA and P(3HO) are missed in this manuscript.
- It would be better if the authors could talk more about the two different types of the vimentin cytoskeleton architecture network “classic” and “nutshell-like”. How these two different types represent cells’ behaviors?
- I think the authors mentioned too many results in their previous work, which may reduce the originality and novelty of this manuscript.
- In this manuscript, the authors compared the VIFs of MEF in PLA, P(3HO) and glass. Why the glass was chosen here? I think normally, cells are cultured in cell cultured dish or plate, so why authors choose the glass instead of cell culture dish in this manuscript?
Reviewer 2 Report
Abstract
- English language requires extensive editing.
- The abstract should be 200 words maximum.
- The abstract should describe the main methods. The authors should include the main findings of this work.
Introduction
- English language requires extensive editing.
- The Introduction should be shortened.
- Reference 6 is incorrectly cited in the text since it does not deal with the PLA topic.
- If Figure 1 is a previous result, must appear explicit in the text and the source from which the information was extracted must be cited.
- If Figure 1 is part of the results of the present manuscript, it must be in the Results Section.It should not appear of the Introduction.
- The type of microscopy used should appear in the figure caption (Figure 1).
Results and Discussion
- Results and Discussion should appear as independent Sections.
- The first three paragraphs of the Results Section (Line 161-176) describe results without displaying them.Figures or tables containing these results must be cited in the text.
- Line 205-211. Authors must specify that these are previous results.
- Some Figures have not a short explanatory title.
- The figure that appears on the line 321 must be named Figure 7 and must be cited on the line 319.
- English language requires extensive editing.
Conclusions
- The conclusions should be shorter and more concise.
References
- The references must be adapted to the style of the International Journal of Molecular Science.
- 29 must be correctly written. Line 514.
- 32 Add the n to the end of Goldman. Line 522
- 34 Line 527. Please check the URL.
- 41 must be completed.
Materials and Methods
- Units of measure must appear separate from the numeric value and must not be enclosed in square brackets.
- Substitute te for the in line 401.
- Secondary antibody for staining should be specified.
General suggestions
- English language requires extensive editing.
- Remove unnecessary spaces between words. Lines: 38, 39, 59, 95, 214, 247, 328, 408, 410, 514, 515,
- Line 53: remove the comma.
- Delete the period before the word dependent, and add the period after the cite [28]. Line 113
- Delete the repeated word (in). Line 143.
- Eliminate the unnecessary period and space. Line 317
